# Novel prognostic determinants of COVID-19-related mortality: A pilot study on severely-ill patients in Russia

Kseniya Rubina[1]*, Anna Shmakova[2], Aslan Shabanov[3], Yulii Andreev[3], Natalia Borovkova[3], Vladimir Kulabukhov[3], Anatoliy Evseev[3], Konstantin Popugaev[3], Sergey Petrikov[3], Ekaterina Semina[2,4]

1 Laboratory of Morphogenesis and Tissue Reparation, Faculty of Medicine, Lomonosov Moscow State University, Moscow, Russia, 2 Laboratory of Molecular Endocrinology, Institute of Experimental Cardiology, Federal State Budgetary Organization National Cardiology Research Center Ministry of Health of the Russian Federation, Moscow, Russia, 3 Department of Intensive Care, Sklifosovsky Research Institute of Emergency Medicine of the Moscow Healthcare Department, Moscow, Russia, 4 Department of Biochemistry and Molecular Medicine, Faculty of Medicine, Lomonosov Moscow State University, Moscow, Russia

* kseniiarubina1971@gmail.com

**Data Availability Statement:** The full data table with all the clinical and biochemical anonymous results is available from the OSF database DOI: 10.

## Abstract

COVID-19 pandemic has posed a severe healthcare challenge calling for an integrated approach in determining the clues for early non-invasive diagnostics of the potentially severe cases and efficient patient stratification. Here we analyze the clinical, laboratory and CT scan characteristics associated with high risk of COVID-19-related death outcome in the cohort of severely-ill patients in Russia. The data obtained reveal that elevated dead lymphocyte counts, decreased early apoptotic lymphocytes, decreased CD14+/HLA-Dr+ monocytes, increased expression of JNK in PBMCs, elevated IL-17 and decreased PAI-1 serum levels are associated with a high risk of COVID-19-related mortality thus suggesting them to be new prognostic factors. This set of determinants could be used as early predictors of potentially severe course of COVID-19 for trials of prevention or timely treatment.

## Introduction

The ongoing wave of COVID-19 infections resulting from an outbreak, which initially started in Wuhan in China, has reached alarming proportions across the globe, as reported by the World Health Organization (WHO) [1]. As of today, no specific antiviral treatment is available for COVID-19. The unprecedented challenge with SARS-CoV-2 infection calls for an integrated approach, encompassing the development of effective vaccines to prevent the spread of the disease, search for new drugs aimed at lowering the viral load and preventing the complications, and quest for the clues to early diagnostics of a severe course of the disease.

The clinical spectrum of COVID-19 manifestations appears to be remarkably wide, ranging from asymptomatic infection, mild upper respiratory tract illness to severe viral pneumonia with respiratory failure, systemic inflammation and immune dysregulation termed "cytokine storm", disseminated intravascular coagulation, multiorgan failure and even death [2–4].

17605/OSF.IO/TZN89 (https://osf.io/tzn89/?view_only=2c70847e06a34fb0bffb9f61246c60db).

**Funding:** The reported study was funded by Russian Foundation for Basic Research (https://www.rfbr.ru/rffi/eng) according to the research project  20-24-60029. The funders had no role in study design, data collection and analysis, decision to publish, or preparation of the manuscript.

**Competing interests:** The authors have declared that no competing interests exist.

Older age, diabetes and cardiovascular comorbidities are among the highest risk factors influencing the disease severity and the clinical outcome of SARS-CoV-2 infection [4, 5]. However, severe COVID-19 has been registered not only in elderly patients or patients suffering from comorbidities, but also in apparently healthy young people [6, 7]. The development of new prognostic algorithms for differential diagnostics to predict and reduce the severity and complications of COVID-19 remains essential.

The data obtained in the present study reveal novel prognostic factors associated with high risk of COVID-19-related death overtime in patients with severe COVID-19. These include: the elevated dead lymphocyte counts, decreased early apoptosis of lymphocytes, decreased CD14+, HLA-Dr+ (human leukocyte antigen D related) monocytes, increased JNK (c-Jun N-terminal kinase) expression in PBMCs (peripheral blood mononuclear cell), increased IL-17 and decreased PAI-1 (plasminogen activator inhibitor-1) serum level. The prognostic value of each determinant can potentially impact decision-making on the level of healthcare and require more extensive follow-up strategies.

To the best our knowledge this is the first integrated study based on the Russian population aimed to reveal the original prognostic COVID-19-related death factors.

## Materials and methods

### Study design and clinical workflow

The present study was a single-center retrospective study carried out on a cohort of 52 symptomatic adult patients with the confirmed SARS-CoV-2 infection, admitted to intensive care unit (ICU) of N.V. Sklifosovsky Research Institute for Emergency Medicine, Moscow from April to July 2020. Reasons for ICU admission included the functional state deterioration with symptoms such as fever, cough, sore throat, signs of pneumonia and respiratory distress in case a patient had been hospitalized from home (51 patients) or transferred from another hospital (1 patient).

Demographic, clinical, and laboratory data were recorded at admission. The study was approved by the ethics committee of Sklifosovsky Research Institute of Emergency Medicine of the Moscow Healthcare Department (protocol #5–120, issued on 01.04.2020). The need for written patient consent was waived by the ethics committee of the hospital because laboratory investigations were conducted according to the local standard. The local ethical committee approved the retrospective study.

Nasopharyngeal swabs were collected from all patients, followed by RT-PCR assay to confirm the SARS-CoV-2 infection. Philips Ingenuity Core 128 CT Scanner (Netherlands) was used for all examinations. In all cases, semi-quantitative CT scoring (severity ranging from 0 to 4) was evaluated according to the protocol recommended by Moscow Healthcare Department adapted from the International Protocols and enriched with local experience [8]. The applied treatments included antibiotics (azithromycin, imipenem/cilastatin, ceftriaxone, ceftriaxone/sulbactame, meropenem/linezolid, levofloxacin, ertapenem, vancomycin), hydroxychloroquine, tocilizumab, convalescent plasma, antivirals (lopinavir, ritonavir, ganciclovir, ribavirin), antihypertensive, anticoagulation, analgesic, NSAID, antiulcer treatment, vitamins and minerals (Supradyn, ascorbic acid, thiamine). Routine blood tests were carried out for all patients and the following parameters were evaluated: white blood cells (WBC), lymphocytes, monocytes, neutrophil, red blood cell (RBC), platelet counts, levels of hemoglobin, serum total protein, albumins, globulins, bilirubin, urea, creatinine, alanine aminotransferase (ALT), aspartate aminotransferase (AST), lactate dehydrogenase (LDH), creatine kinase (CK), prothrombin, fibrinogen, D-dimer levels, activated partial thromboplastin time (APTT) and the international normalized ratio (INR).

## Biochemical parameters analysis in blood serum and PBMCs

In addition to blood tests routinely and periodically performed in hospital, we assessed the levels of plasminogen (PLG), plasminogen activator inhibitor 1 (PAI-1), IL-1α, IL-17, transforming growth factor beta (TGFβ), tumor necrosis factor alpha (TNFα), and adiponectin (ADP) in blood serum. Blood serum was collected and immediately frozen. Commercially available ELISA kits (SEA605Hu for ADP, SEA071Hu for IL-1α, SEA063Hu for IL-17, SEA532Hu for PAI-1, SEB236Hu for PLG, SEA124Hu for TGFβ, and SEA133Hu for TNFα (Cloud-Clone Corp., Wuhan, Hubei) were used to evaluate the corresponding analytes according to the manufacturers' protocols. The 3,3', 5,5'-tetramethylbenzidine (TMB) was used as a chromogen and the maximum absorption wavelength of 450 nm was detected using a microplate reader Multiscan Ascent (Thermo Fisher). The experiments were performed in duplicate.

Whole blood specimens were collected into EDTA-containing tubes either directly labelled with antibodies or subjected to PBMC purification using Lympholyte-H (Cedarlane) density gradient centrifugation. Antibodies against the following surface markers were applied for whole blood sample labelling: CD95 (mouse monoclonal anti-hu CD95-FITC 1F-362-T100, EXBIO), CD14 (mouse monoclonal anti-hu CD14-PE Conjugated Antibody A07764, Beckman Coulter) and HLA-Dr (HLA-Dr-FITC Conjugated Antibody, IM1638U, Beckman Coulter). Isotypic non-immune IgG (IgG1 Mouse-FITC Isotype Control, A07795, Beckman Coulter) was used as a control. For lymphocyte death analysis, PBMCs were stained with anti-CD45, Annexin V, 7AAD (mouse monoclonal anti-hu CD45-FITC Conjugated Antibody A07782 Beckman Coulter; Annexin V-FITC Kit-AAD Kit, IM3614, Beckman Coulter; 7-AAD Viability Dye A07704, Beckman Coulter). Labelled cells were analysed by flow cytometry with CYTOMICS FC500 flow cytometer (Beckman Coulter) and CXP software. Lymphocyte and monocyte populations were gated on FSC vs. SSC plot. The following parameters were assessed: the absolute number per 1 μl and the percentage of dead lymphocytes (CD45 +/7AAD+), the percentage of early (Annexin V+/7AAD) and late apoptotic (Annexin V +/7AAD+) lymphocytes as well as the percentage of CD95+ lymphocytes and HLA-Dr+ monocytes (CD14+/HLA-Dr+). The gating strategy is shown in **S1 Fig**.

PBMC lysates were prepared using protease inhibitor Gordox®, and the apoptotic marker expression, such as Akt, Bad, Bcl-2, Caspase-8, Caspase-9, JNK and p53 was evaluated using Milliplex®MAP (Early Apoptosis 7-plex Magnetic Bead Kit, cat#48-660MAG) assay and Luminex® (USA). The values for each parameter are presented as Median Fluorescence Intensity (MFI). The experiments were performed in duplicate.

## Outcome and exposures

The outcome used in the survival analysis in the current study was COVID-19-related death, defined on the basis of International Classification of Diseases 10 code U07.1 (COVID-19, virus identified), recorded as either a primary underlying or secondary cause of death.

With respect to potential risk factors assessed as exposures, we obtained data for sex, age, body mass index (BMI), tobacco smoking status, chest CT severity score (the worst CT imaging data per patient presented in **Table 1**). Patients were grouped by age as: younger than 40 years, 40–49 years, 50–59 years, 60–69 years, 70–79 years, and 80 years or older. BMI was grouped as: less than 30 (no evidence of obesity), 30–34.9 (obese class I), 35–39.9 (obese class II), and 40 kg·m$^{-2}$ or higher (obese class III). Tobacco smoking status was identified as smoker, non-smoker (not a current smoker), or missing data. Chest CT score was grouped as 1–2 or 3–4 based on semi-quantitative CT scoring (ranging from 0 to 4) evaluated according to the protocol recommended by Moscow Healthcare Department adapted from the International Protocols and enriched with local experience [8].

**Table 1. Cohort description with COVID-19 deaths by patient characteristics.**

| | All patients (n = 52) | Alive (n = 41) | Deaths (n = 11) | P |
|---|---|---|---|---|
| **Sex** | | | | |
| Male | 28 (54%) | 23 (56%) | 5 (45%) | 0.5295 |
| Female | 24 (46%) | 18 (44%) | 6 (55%) | |
| **Age, years** | | | | |
| 18–39 | 3 (6%) | 3 (7%) | 0 (0%) | 0.2451 |
| 40–49 | 8 (15%) | 7 (17%) | 1 (9%) | |
| 50–59 | 12 (23%) | 11 (27%) | 1 (9%) | |
| 60–69 | 11 (21%) | 8 (19%) | 3 (27%) | |
| 70–79 | 7 (13%) | 6 (15%) | 1 (9%) | |
| 80+ | 11 (21%) | 6 (15%) | 5 (45%) | |
| **BMI, kg·m$^{-2}$** | | | | |
| <30 | 32 (62%) | 25 (61%) | 7 (64%) | 0.7470 |
| 30–34.9 | 16 (31%) | 12 (29%) | 4 (36%) | |
| 35–39.9 | 2 (4%) | 2 (5%) | 0 (0%) | |
| 40+ | 2 (4%) | 2 (5%) | 0 (0%) | |
| **Tobacco smoking status** | | | | |
| Non-smoker | 41 (79%) | 35 (85%) | 6 (55%) | 0.5597 |
| Smoker | 2 (4%) | 2 (5%) | 0 (0%) | |
| Missing data | 9 (17%) | 4 (10%) | 5 (45%) | |
| **Chest CT severity score (the worst CT imaging data per patient)** | | | | |
| 0–2 | 32 (62%) | 30 (73%) | 2 (18%) | 0.0009 *** |
| 3–4 | 20 (38%) | 11 (27%) | 9 (82%) | |
| **Major treatment applied[a]** | | | | |
| Azithromycin | 33 (64%) | 27 (66%) | 6 (55%) | 0.4892 |
| Other antibiotics | 23 (44%) | 12 (29%) | 11 (90%) | <0.0001**** |
| Hydroxychloroquine | 46 (88%) | 38 (93%) | 8 (73%) | 0.0599 |
| Tocilizumab | 14 (27%) | 12 (29%) | 2 (18%) | 0.4617 |
| Convalescent plasma | 17 (33%) | 16 (39%) | 1 (9%) | 0.0602 |
| Antivirals | 11 (21%) | 11 (27%) | 0 (0%) | 0.0530 |
| Vitamins + minerals | 38 (73%) | 31 (76%) | 7 (64%) | 0.4266 |
| Antihypertensive | 38 (73%) | 31 (76%) | 7 (64%) | 0.4266 |
| Anticoagulation | 49 (94%) | 39 (95%) | 10 (82%) | 0.5946 |
| Antiulcer | 43 (83%) | 32 (78%) | 11 (100%) | 0.0875 |
| **Underlying comorbidities[b]** | | | | |
| Hypertension | 28 (54%) | 22 (54%) | 6 (55%) | 0.9582 |
| Congestive heart failure | 8 (15%) | 8 (20%) | 0 (0%) | 0.1112 |
| Ischemic heart disease | 9 (17%) | 7 (17%) | 2 (18%) | 0.1545 |
| Chronic kidney disease | 4 (8%) | 2 (5%) | 2 (18%) | 0.1415 |
| Diabetes | 3 (6%) | 3 (7%) | 0 (0%) | 0.3554 |
| Malignancy | 5 (10%) | 3 (7%) | 2 (18%) | 0.2778 |
| Cerebrovascular disease | 4 (8%) | 2 (5%) | 2 (18%) | 0.1415 |
| **Complications during hospital stay[c]** | | | | |
| Hydrothorax | 6 (12%) | 1 (2%) | 5 (45%) | <0.0001**** |
| Intoxication, sepsis, multiple organ failure | 8 (15%) | 0 (0%) | 8 (73%) | <0.0001**** |

*(Continued)*

**Table 1.** (Continued)

|  | All patients (n = 52) | Alive (n = 41) | Deaths (n = 11) | *P* |
|---|---|---|---|---|
| Acute respiratory distress syndrome | 4 (8%) | 0 (0%) | 4 (36%) | <0.0001**** |
| Hemorrhagic shock | 1 (2%) | 0 (0%) | 1 (9%) | 0.0512 |
| Acute myopericarditis | 1 (2%) | 1 (2%) | 0 (0%) | 0.6054 |
| Pseudomembranous colitis | 1 (2%) | 1 (2%) | 0 (0%) | 0.6054 |
| Deep vein thrombosis of the lower extremities | 3 (6%) | 1 (2%) | 2 (18%) | 0.0468* |

Data are presented as n (%). BMI = body mass index. *P* values of $\chi^2$ test comparison of different factors between alive and deaths groups are provided.

***–$P < 0.001$

****–$P < 0.0001$.

a–*P* value provided for the comparison of each treatment *vs* non-treatment.

b–*P* value provided for the comparison of each comorbidity *vs* no such comorbidity.

c–*P* value provided for the comparison of each complication *vs* no such complication.

Routine blood test parameters were organized into three groups relying on the pre-specified reference range accepted in Moscow hospital laboratories, including the clinical laboratory of N.V. Sklifosovsky Research Institute for Emergency Medicine: lower, within the reference range or higher than the reference range. The accepted reference values of the blood tests parameters are presented in **S1 Table**. In case the data of several measurements were available, the mean of all measurements was calculated and used for subsequent analysis. To ensure an adequate repartition in the groups, the categorization of the biochemical parameters in the blood serum and leukocytes was performed basing on the pre-specified reference values provided by the manufacturer in the kit instructions, literature data and evaluation of the histogram distribution data. The cut-off values for biochemical serum and leukocyte parameters are presented in **S2 Table.**

## Statistical analysis

Categorical variables are described as numbers and percentages. Continuous variables are reported as the median (interquartile range). The contribution of varioust factors to the difference between alive and non-survival groups were assessed using $\chi^2$ test. To identify risk factors associated with COVID-19-related death, a survival analysis of the outcome was performed: time was defined as the time from hospital admission to the end of a follow-up (hospital discharge or death) and the event defined as death or hospital discharge after recovery. The Kaplan-Meier method was used to analyze cumulative survival curves and their difference between the groups categorized by type of exposure compared by log-rank tests. Exposures with difference between the groups $P < 0.05$ by log-rank tests were considered significant. We created univariable Cox proportional hazards survival analysis models for the evaluated parameters of each patient to test for their association with survival. We also created multivariable Cox proportional hazards survival analysis models to identify independent risk factors. Complete cases were considered. The inclusion of variables in the final models were based on the following criteria: 1) significance in Kaplan-Meier and/or univariable analyses; 2) association with survival according to the previous studies [9–14]; 3) clinical knowledge.

Considering the small size of the cohort and to avoid overfitting in the model, we created three separate final multivariable Cox models, with variables grouped as clinical characteristics and hematological results; clinical biochemical routine blood test results; laboratory blood findings. The data were analyzed in R studio v.1.2.5042 with the survival and survminer packages.

## Results

A total of 52 patients, aged 26–88, with confirmed SARS-CoV-2 infection were included in the study cohort (**Table 1**). The median age was 64 (54–76) years; the median body mass index (BMI) was 28.7 (25.8–31.1) kg·m$^{-2}$; 28 patients (54%) were men. 1 patient was admitted to intensive care unit (ICU) with CT score 0; 13 patients with CT score 1; 18 patients with CT score 2; 13 patients with CT score 3, and 4 patients with CT score 4. Overall, 11 (21%) patients died. The group of non-survival patients didn't differ significantly from alive patients by sex, age, BMI, tobacco smoking status as compared by $\chi^2$ test (**Table 1**). No difference has been detected in the applied treatment regimens, except a more frequent usage of additional antibiotic therapy in non-survival patients presumably due to their insensitivity to azithromycin, as well as no significant difference in the frequency of underlying comorbidities was noted (**Table 1**). Chest CT score was significantly worse in the group of non-survival patients ($P = 0.0009$, $\chi^2$ test). Patients from non-survival group had much more severe complications during hospital stay, including hydrothorax ($P < 0.0001$, $\chi^2$ test), intoxication, sepsis, multiple organ failure ($P < 0.0001$, $\chi^2$ test), acute respiratory distress syndrome ($P < 0.0001$, $\chi^2$ test) (**Table 1**).

To identify prognostic determinants of COVID-19-related mortality, the survival analysis was performed with time defined as a period from hospital admission to the end of the follow-up (hospital discharge or death) and event defined as death or hospital discharge after recovery (**Fig 1**). The Kaplan-Meier survival function analysis revealed that 20-day survival probability

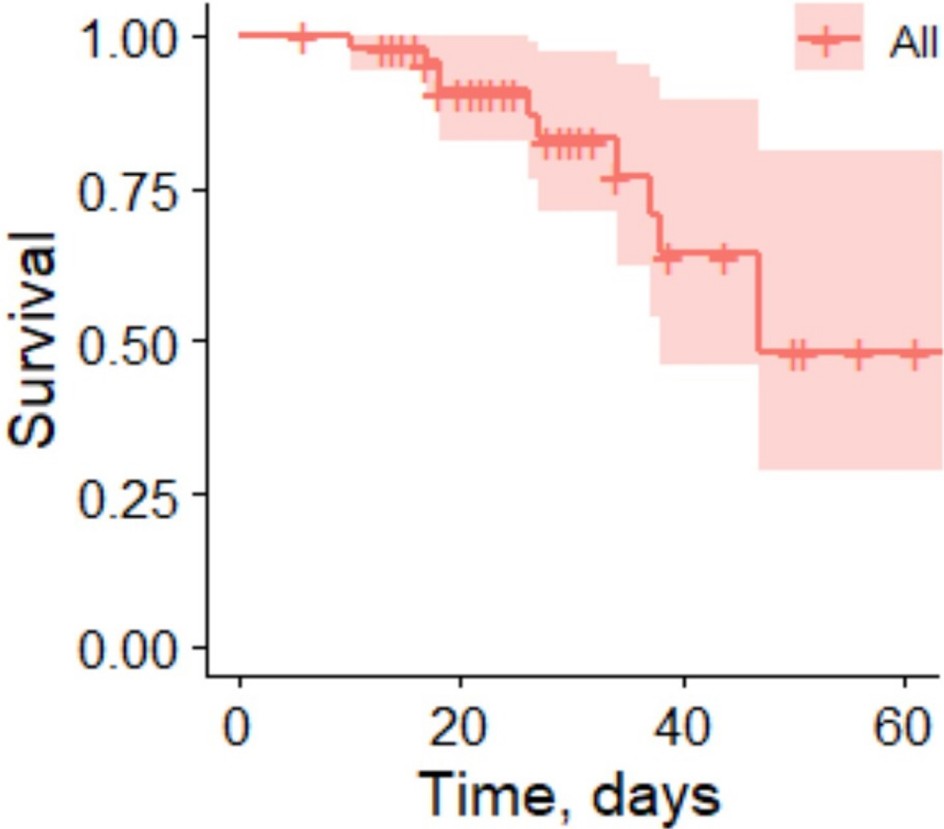

**Fig 1. Kaplan-Meier survival curve with associated 95% confidence intervals (shaded area) for patients admitted with COVID-19 infection to ICU of N.V. Sklifosovsky Research Institute for Emergency Medicine, Moscow from April to July 2020.** The fraction of survival is expressed on the y-axis, while time (days) of the observation period is expressed on the x-axis. Vertical traits indicate censored data (hospital discharge).

was 91.0% (95% CI 82.9–99.9%), 40-day survival probability was 64.3% (95% CI 46.1–89.5), 60-day survival probability was 48.2% (95% CI 28.7–81.0%).

We performed the Kaplan-Meier analysis to compare the survival over time between different groups of patients for each variable. The Kaplan-Meier survival plots for the clinical and routine blood test factors with statistically significant results are presented in **Fig 2**. We found, that the risk of COVID-19-related death over time was significantly increased in patients with CT score 3–4 at admission ($P = 0.042$), patients on pulmonary ventilation ($P = 0.00047$), patients with white blood cell (WBC) count more than $11.8 \cdot 10^3/\mu l$ ($P = 0.001$), neutrophil count more than $8.2 \cdot 10^3/\mu l$ ($P = 0.00027$), lymphocyte count less than $1.1 \cdot 10^3/\mu l$ ($P = 0.018$), increased serum levels of globulins more than 30 g/l ($P = 0.046$), urea more than 7.2 mM ($P = 0.01$), creatine kinase (CK) more than 200 U/l ($P = 0.023$), prothrombin less than 78% ($P = 0.023$), the international normalized ratio (INR) more than 1.17 ($P = 0.02$) and D-dimer more than 2 mg/l ($P = 0.0078$). Gender, age, BMI, tobacco smoking status and other parameters of routine blood tests were not associated with statistically significant differences as revealed by Kaplan-Meier analysis (**S2 Fig**).

The Kaplan-Meier survival plots for the biochemical test factors being statistically significant are presented in **Fig 3**. We found, that the risk of COVID-19-related death over time was significantly increased in patients with more than 100 dead lymphocytes in 1 μl ($P = 0.0026$), less than 5% of early apoptotic lymphocytes ($P = 0.0081$), less than 85% of CD14+/HLA-Dr + monocytes ($P = 0.029$), increased expression of JNK in PBMCs (>200 MFI, $P = 0.034$), increased serum levels of IL-17 (>7 pg/ml, $P = 0.037$) and decreased PAI-1 serum levels (< 40 pg/ml, $P = 0.00073$). Percentage of dead and late apoptotic lymphocytes, percentage of CD95 + lymphocytes, expression of Akt, Bad, Bcl-2, Caspase-8, Caspase-9, p53 and serum levels of plasminogen (PLG), IL-1α, TGFβ, TNFα, adiponectin (ADP) were not associated with statistically significant differences in survival as revealed by Kaplan-Meier analysis (**S3 Fig**).

To identify the independent risk factors, we created multivariable Cox proportional hazards survival analysis models separately for clinical characteristics and hematology results; for clinical biochemical routine blood test results and for laboratory blood findings (**Table 2**). On multivariable Cox proportional hazards survival analysis of clinical and routine hematological test parameters, ventilation (hazard ratio, HR 55.66 (2.2–1388.4), $P = 0.014$) and platelet count (HR 1.01 (1.002–1.027), $P = 0.017$) were found to be the independent prognostic factor for COVID-19 related death. On multivariable analysis of clinical biochemical routine blood test results, the serum levels of globulins (HR 1.32 (1.03–1.70), $P = 0.026$), CK (HR 1.02 (1.005–1.04), $P = 0.013$) and D-dimer (3.41 (1.11–10.50), $P = 0.033$) were found to be the independent prognostic factor for COVID-19 related death. On multivariable analysis of laboratory blood findings, the serum level of IL-17 (HR 1.07 (1.001–1.14), $P = 0.046$) and PAI-1 (HR 0.82 (0.69–0.98), $P = 0.026$) were found to be the independent prognostic factor for COVID-19 related death. The results of univariable Cox proportional hazards survival analysis for each parameter are presented in **S3 Table**.

## Discussion

With the rapid spread of COVID-19 around the world since its outbreak in early 2020, it has stirred up an international concern and still remains so. Moreover, the poor survival rates of high-risk patients warrant investigation into novel prognostic factors implicated in COVID-19 pathogenesis, as well as diagnostic and treatment options aiming to reduce COVID-19-related burden in healthcare system. To address this question, we carried out a single-center cohort study on 52 symptomatic adult patients with the confirmed SARS-CoV-2 infection, admitted to ICU of N.V. Sklifosovsky Research Institute for Emergency Medicine, Moscow from April

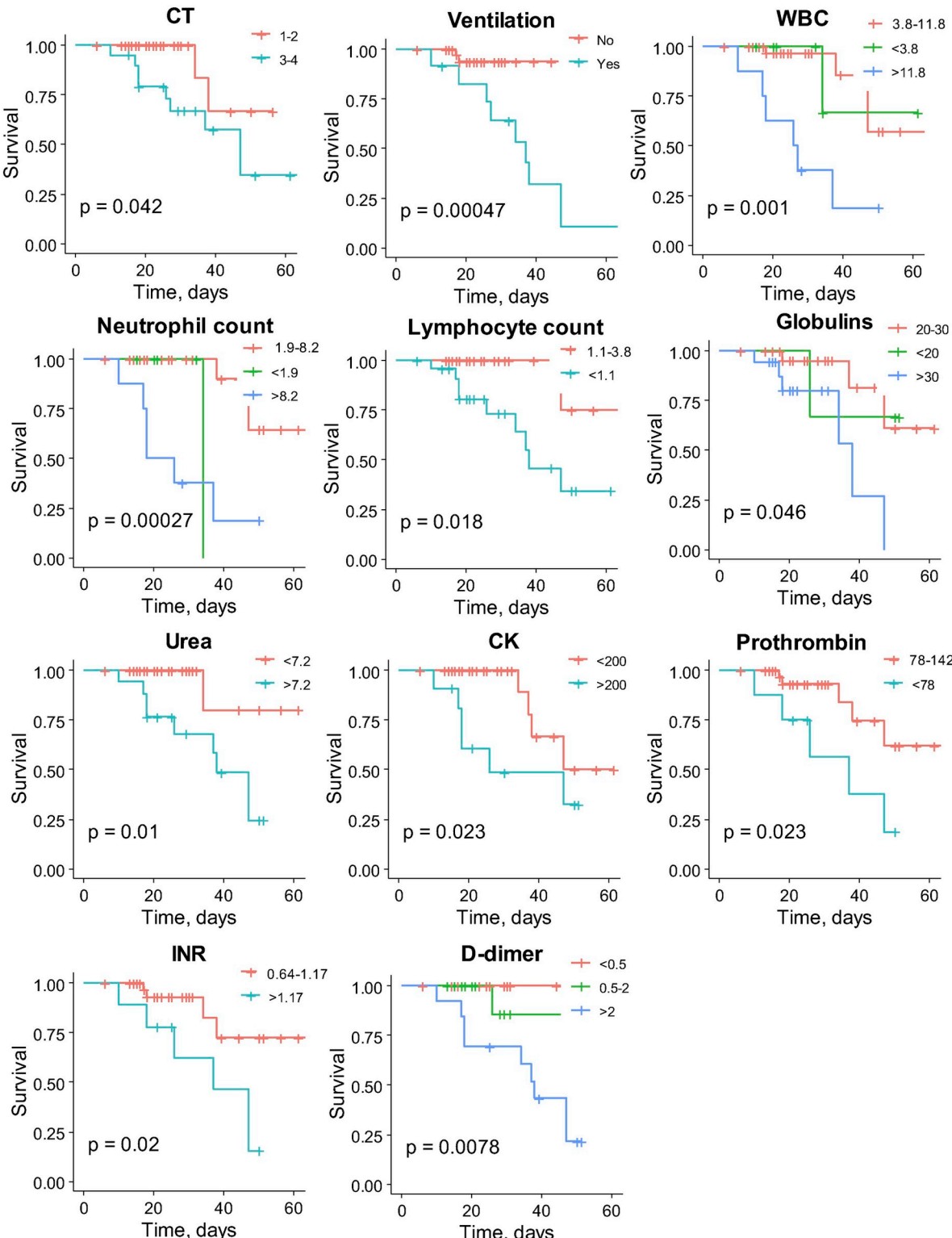

**Fig 2. Kaplan-Meier survival curves for patients grouped by clinical and routine blood test parameters.** The fraction of survival is expressed on the y-axis, while time (days) of the observation period is expressed on the x-axis. Vertical traits indicate censored data (hospital discharge). *P* values of log-rank tests are indicated for each graph. Reference group is shown in red.

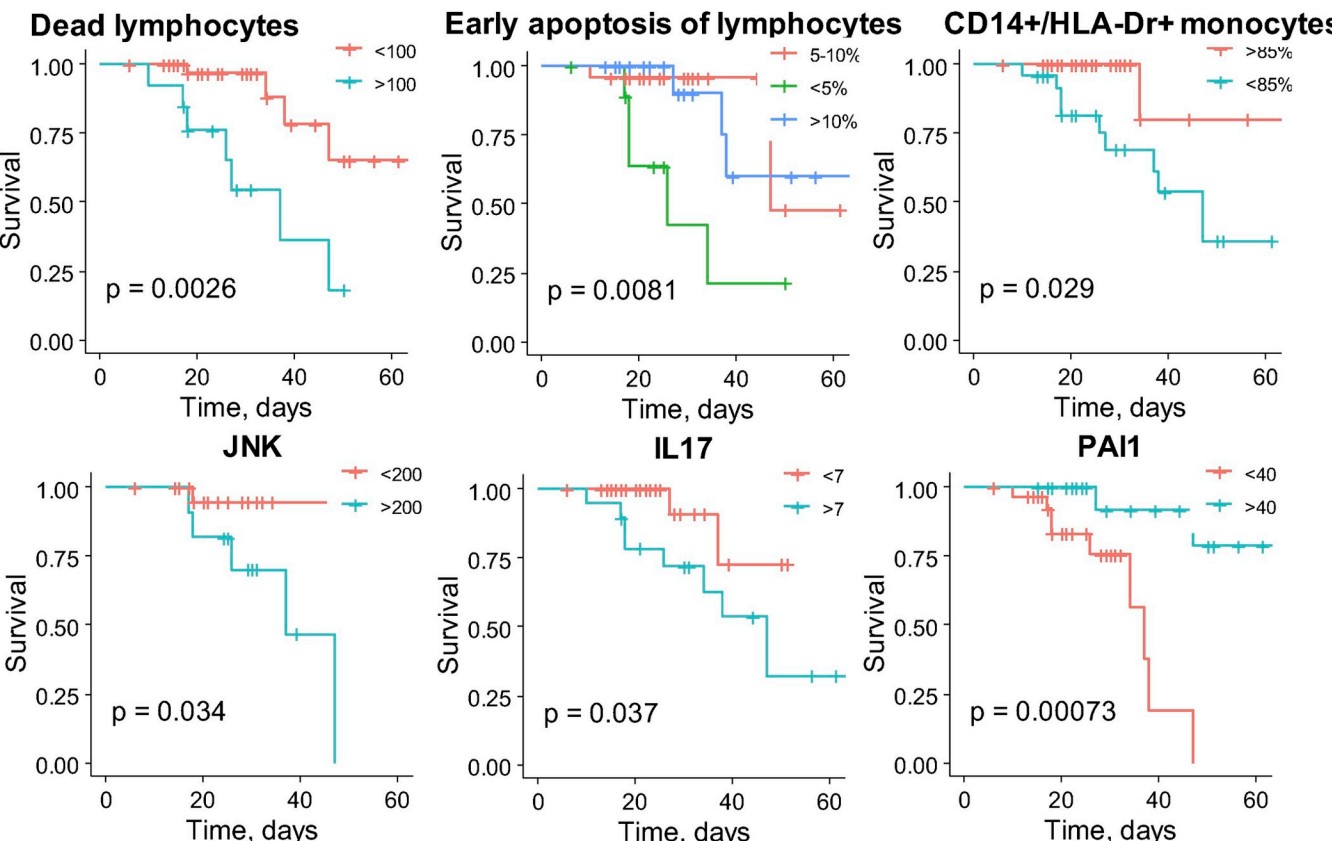

**Fig 3. Kaplan-Meier survival curves for patients grouped by biochemical test parameters.** The fraction of survival is expressed on the y-axis, while time (days) of the observation period is expressed on the x-axis. Vertical traits indicate censored data (hospital discharge). *P* values of log-rank tests are indicated for each graph. Reference group is shown in red.

to July 2020. The overall ICU mortality in this cohort was 21%, which is consistent with previous reports [15, 16]. Patients from non-survival group had much more severe complications during the hospital stay, including hydrothorax, intoxication, sepsis, multiple organ failure, acute respiratory distress syndrome, which led them eventually to death (**Table 1**).

To identify prognostic determinants of COVID-19-related mortality, we first assessed classical clinical and routine blood test parameters (**Fig 2 and S2 Fig**). Chest CT imaging has been recognized to play a pivotal role in monitoring COVID-19 disease progression and predicting adverse prognosis. We found that CT score 3–4 at admission and pulmonary ventilation significantly increase the risk of COVID-19-related death over time (**Fig 2**) that is in accordance with the previously published data [2, 17–22].

A compelling body of evidence indicate that clinical and routine blood test factors such as an increased WBC count [2, 23–30], elevated neutrophil count [2, 24–31], low lymphocyte count [2, 4, 17, 25, 26, 28–30] are statistically significant and strongly correlate with the adverse outcome in COVID-19 patients. Previously, neutrophil-lymphocyte ratio (NLR) was shown to be a prognostic marker of systemic inflammation in various pathological conditions [32, 33]. Moreover, NLR was found to have a high prognostic value in COVID-19 patients with the increase in neutrophils and the decrease in the total number of lymphocytes in the peripheral blood. Generally, the COVID-19 patients with increased NLR have a poor prognosis and a high risk of death [34]. Our data are in line with recently published papers: we found that the risk of COVID-19-related death over time was significantly increased in patients with elevated

**Table 2. Multivariable Cox proportional hazards survival analysis models for 3 groups of prognostic factors.**

| Parameter | HR (95% CI) | P value |
|---|---|---|
| **Clinical characteristics and hematology results** | | |
| CT | 0–2: reference 3–4: 0.52 (0.03–7.85) | 0.634 |
| Ventilation | No: reference Yes: 55.66 (2.23–1388.40) | 0.014 * |
| WBC, $10^3$/μl | 0.58 (0.11–2.96) | 0.509 |
| Neutrophil count, $10^3$/μl | 2.68 (0.57–12.48) | 0.210 |
| Lymphocyte count, $10^3$/μl | 0.29 (0.01–6.93) | 0.447 |
| Monocyte count, $10^3$/μl | 0.15 (0.00–1564.4) | 0.692 |
| Platelet count, $10^3$/μl | 1.01 (1.002–1.027) | 0.017 * |
| **Clinical biochemical routine blood test results** | | |
| Globulins, g/l | 1.32 (1.03–1.70) | 0.026 * |
| Urea, mM | 0.88 (0.62–1.20) | 0.467 |
| Creatinine, μM | 1.02 (0.99–1.10) | 0.244 |
| ALT, U/l | 0.99 (0.93–1.00) | 0.638 |
| AST, U/l | 1.01 (0.98–1.00) | 0.572 |
| LDH, U/l | 0.99 (0.985–1.00) | 0.705 |
| CK, U/l | 1.02 (1.005–1.04) | 0.013 * |
| D-dimer, mg/l | 3.41 (1.11–10.50) | 0.033 * |
| **Laboratory blood findings** | | |
| Dead lymphocyte count | 1.02 (1.00–1.04) | 0.054 |
| Early lymphocyte apoptosis | 1.09 (0.77–1.55) | 0.624 |
| CD14+, HLA-Dr+ monocytes, % | 0.96 (0.91–1.01) | 0.108 |
| IL-17, pg/ml | 1.07 (1.001–1.14) | 0.046 * |
| IL-1α, pg/ml | 1.09 (0.90–1.31) | 0.385 |
| PLG, μg/ml | 0.66 (0.32–1.32) | 0.238 |
| PAI-1, pg/ml | 0.82 (0.69–0.98) | 0.026 * |
| TNFα, pg/ml | 0.90 (0.68–1.19) | 0.464 |
| TGFβ, pg/ml | 1.00 (1.00–2.02) | 0.069 |
| ADP, μg/ml | 0.56 (0.30–1.08) | 0.083 |

HR–hazard ratio, CI–confidence interval, ALT–alanine aminotransferase, AST–aspartate aminotransferase, LDH—
lactate dehydrogenase (LDH), CK–creatine kinase, PLG–plasminogen, PAI-1 –plasminogen activator inhibitor 1,
ADP—adiponectin.

*–$P < 0.05$.

WBC and neutrophil count, and decreased lymphocyte count (**Fig 2**). However, when considering clinical characteristics and hematology test results together, only ventilation and elevated platelet count were found to be independent risk factors for mortality (**Table 2**).

Clinical biochemical routine blood test results, such as increased urea [2, 23, 27, 29–31, 35], upregulated CK [2, 4, 17, 26, 30], and augmented level of serum globulins [31] or immunoglobulins [36] can be also valuable in predicting the mortality in COVID-19 patients. Elevated plasma D-dimer is a well-established prognostic factor for adverse outcome in respiratory diseases and COVID-19 [34, 37, 38]. Our results go along with these previously published data and ascertain that the COVID-19-related death over time is significantly increased in patients with elevated serum levels of globulins, urea, CK, augmented D-dimer and INR, and decreased prothrombin (**Fig 2**). CK, globulins, and D-dimer were identified as independent risk factors for mortality among the tested routine clinical biochemical results (**Table 2**).

A potentially practical prognostic opportunity was proposed by other cellular and biochemical factors, that are not in routine clinical practice, such as the sustained elevated production of IL-6 [2, 4, 17, 23, 24, 26, 27, 29–31, 39], IL-1, IL-2R, IL-8, IL-10 [2, 17, 40] and some other blood circulating factors, whose early evaluation anticipated disease progression and duration of hospital stay of COVID-19 patients [41]. Such interleukins as IL-1β, IL-6, IL-17A, TNFα, and monocyte chemoattractant peptide (MCP)-1 were identified in patients with severe COVID-19 [42]. Huang et al. [43] demonstrated the upregulated levels of IL-1β, IL-6, IL-8, IL-17, interferon γ (IFN-γ), TNFα in patients with COVID-19 as compared to healthy donors. The same study revealed increased plasma concentrations of MCP-1 and TNFα in patients with COVID-19 admitted to the ICU. Similarly, Chen and co-authors [6] reported the correlation between high concentrations of plasma IL-6 and TNFα and the severe course of COVID-19. In contrast, in the study by Kang and co-authors these cytokines, including IL-1β, IL-12p40, and IL-17 were undetectable in patients with severe COVID-19 [44]. Several clinical trials investigating interleukin inhibitors (IL-1 inhibitor anakinra, and IL-6 inhibitors tocilizumab or sarilumab) revealed that IL-1 inhibition, but not IL-6 inhibition, was associated with a significant reduction of mortality in patients admitted to hospital with COVID-19, ARDS (acute respiratory distress syndrome), and hyperinflammation [45]. That is why we next accessed the association of different cellular and biochemical factors with patient survival (**Fig 3 and S3 Fig**). In the present study IL-1α, TNFα and TGFβ were not associated with the statistically significant difference in patient survival (**S3 Fig**); however, elevated IL-17 and decreased PAI-1 were determined as significant and independent risk factors for COVID-19 related death (**Fig 3, Table 2**).

IL-17 is a promising target in COVID-19, since it operates upstream of both, IL-1 and IL-6 [46]. Although the pathogenic role of IL-17 has been suggested previously, this cytokine was not detected as a mortality risk factor in COVID-19 patients [42, 46]. MERS-CoV infection was reported to be associated with an increased serum level of IL-17, although its correlation with mortality was not demonstrated [47]. IL-17 polymorphisms are associated with susceptibility and poor prognosis in acute respiratory distress syndrome [48]. Meanwhile, in obese patients with confirmed COVID-19, IL-17 boosts neutrophil recruitment via induction of inflammatory cytokine production by respiratory epithelium, smooth muscle cells and fibroblasts resulting in respiratory dysfunction [42].

PAI-1 is a well-known inhibitor of the plasminogen activators (tissue plasminogen activator–tPA, and urokinase–uPA) and hence the overall fibrinolysis [49]. Of note, PAI-1 levels are increased in hypertension, obesity, diabetes, cardiovascular diseases, and old age, which is consistent with the scenario of COVID-19 disease [50, 51]. Plasmin, and other proteases, may cleave the furin site in the S protein of SARS-CoV-2, thus increasing the virus virulence. Moreover, the dysregulation in fibrinolytic homeostasis in COVID-19 patients with excessive fibrin degradation may result in an unbalanced fibrinolysis inducing multiple organ hemorrhage [52, 53]. Hyperfibrinolysis resulting from excessive plasmin activity is also associated with elevated D-dimer in severely-ill COVID- 19 patients. While Umemura et al. reported an unchanged PAI-1 level in COVID-19 patients [54], several other groups detected the increased levels of PAI-1, tPA and uPA in COVID-19 patients as compared to heathy controls [55, 56]. Zou and co-authors demonstrated that high levels of tPA and PAI-1 were associated with worse respiratory status in patients hospitalized with COVID-19 [50]. It has been found that overproduced PAI-1 binds to specific receptors on macrophages and upregulates the production of proinflammatory cytokines and chemokines, which further activate the innate immune cells within the infected lungs resulting in lung damage. In turn, high hypoxic environment additionally stimulates PAI-1 production creating a vicious cycle of cytokine storm generated during SARS virus pathogenesis [53]. In contrast, our data indicate that COVID-19-related

mortality is associated with decreased PAI-1 (**Fig 3**), establishing that PAI-1 being the constituent component of the fibrinolytic system plays an important role in COVID-19 pathogenesis. The explanation for the reduced PAI-1 content in the critically-ill patients may reside in compensation for the hyperactivation of plasminogen activator system, reflecting the exhaustion of plasmin inhibitors.

Earlier, it has been demonstrated that COVID-19 patients with severe respiratory failure exhibit low HLA-Dr expression on CD14+ monocytes relative to that of mild COVID-19 patients or healthy volunteers [57, 58]. Xu and co-authors hypothesized that monocytes in severe COVID-19 displayed a phenotype similar to immunosuppressive monocytic myeloid-derived suppressor cells (CD14+/HLA-DR-/lo) [58]. Interestingly, the expression of HLA-Dr on monocytes from healthy donors decreased following the cultivation in media with plasma from patients with COVID-19 who have immune dysregulation [59]. We have defined that the decrease in CD14+/HLA-Dr+ monocyte fraction is associated with mortality in COVID-19 patients (**Fig 3**).

We also assessed the association of lymphocyte apoptosis with patient survival. The biological role of lymphocyte apoptosis in infectious disease is ambiguous; whether it operates as an intrinsic protective mechanism or mediates a severe course of the disease still remains unclear. Lymphocytes can undergo extensive and ostensibly out-of-control apoptosis during severe infections, potentially contributing to immunosuppression [60]. In the majority of viral infections, lymphocytes are subjected to apoptosis, which is considered to be a defense mechanism preventing the spread of infection without a local inflammatory reaction [61]. Although lymphocyte count can be initially augmented at the onset of COVID-19, its subsequent decline in the course of infection may result in lymphocytopenia and immunodeficiency associated with an increased COVID-19 severity [62]. The frequency of peripheral blood lymphocyte subpopulations, including T-CD4+, T-CD8+, NK, B cells, and monocytes, and their apoptosis pattern was assessed in Iranian COVID-19 patients. The authors reported a marked increase in apoptosis of mononuclear cells from COVID-19 patients as compared to control, however no association with mortality has been addressed in this study [63]. Our data indicate that elevated dead lymphocyte counts are associated with COVID-19 mortality (**Fig 3**), however, early lymphocyte apoptosis is rather a protective mechanism reflecting the intensity of the adaptive cell immune response to an infectious agent, whereas a decrease in the early apoptosis of lymphocytes is associated with COVID-19-related mortality (**Fig 3**).

The C-Jun NH2-terminal kinase (JNK1/JNK2) pathway activation ultimately leads to various cellular effects such as inflammatory response, cell proliferation, survival or even apoptosis. JNK signaling pathway was suggested to play a prominent role in immune response to viral infection due to activation of several interleukins (IL-2, IL-4) and IFN-γ [64]. To uncover the mechanisms of JNK-mediated inflammation in COVID-19, Shirato and Kizaki carried out an *in vitro* study and analyzed the effects of SARS-CoV-2 spike protein S1 subunit on murine and human macrophages. Exposure to S1 subunit activated pro-inflammatory mediators (nuclear factor-κB (NF-κB) and JNK) signaling pathways via the activation of a toll-like receptor 4 (TLR4) on macrophage surface. Pro-inflammatory cytokine (TNFα, IL-6, IL-1β, and nitric oxide) production induced by S1 was abrogated by specific inhibitors of NF-κB and JNK pathways [65]. In line with the suggested strategy for inhibition of JNK pathway in COVID-19 critically-ill patients as a promising therapeutic approach [66], here we present the first clinical study revealing the increased JNK expression in PBMCs to be a significantly associated with COVID-19-related mortality (**Fig 3**). It is tempting to speculate that the elevated JNK, which has been demonstrated to induce TLR4-mediated apoptosis in various types of cells [65, 67], and the decreased level of lymphocyte early apoptosis in severe COVID-19 patients revealed in the present study are intertwined. However, this assumption needs further investigation.

Towards that end, non-routine biochemical test results determined IL-17 and PAI-1 as independent prognostic factors for COVID-19 related death among others (Table 2), underlying their potential significance as predictors of mortality for patients with COVID-19.

## Conclusions

In conclusion, this study unveiled an integrated set of immunological features and new prognostic markers of patients with severe COVID-19 that are significant predictors of the related mortality. Routine parameters, such as CT score, ventilation, increased WBC and neutrophil count, elevated serum levels of globulins, urea, CK, D-dimer and INR, in parallel with decreased prothrombin and lymphocyte count are risk factors for COVID-19-related mortality. Extended analysis of factors, non-routinely used in clinical practice, suggests a high risk of COVID-19-related death overtime in patients with elevated dead lymphocyte counts, decreased early apoptosis of lymphocytes, reduced fraction of CD14+/HLA-Dr+ monocytes, increased JNK expression in PBMCs, elevated IL-17 and decreased PAI-1 serum levels (Fig 4). In multivariable comparison, IL-17 and PAI-1 are independent prognostic factors among others. Our analysis revealed novel determinants of a high COVID-19-related death risk overtime that can provide for patient stratification at admission to ICU or point to the potential requirement of a more extensive follow up treatment strategies.

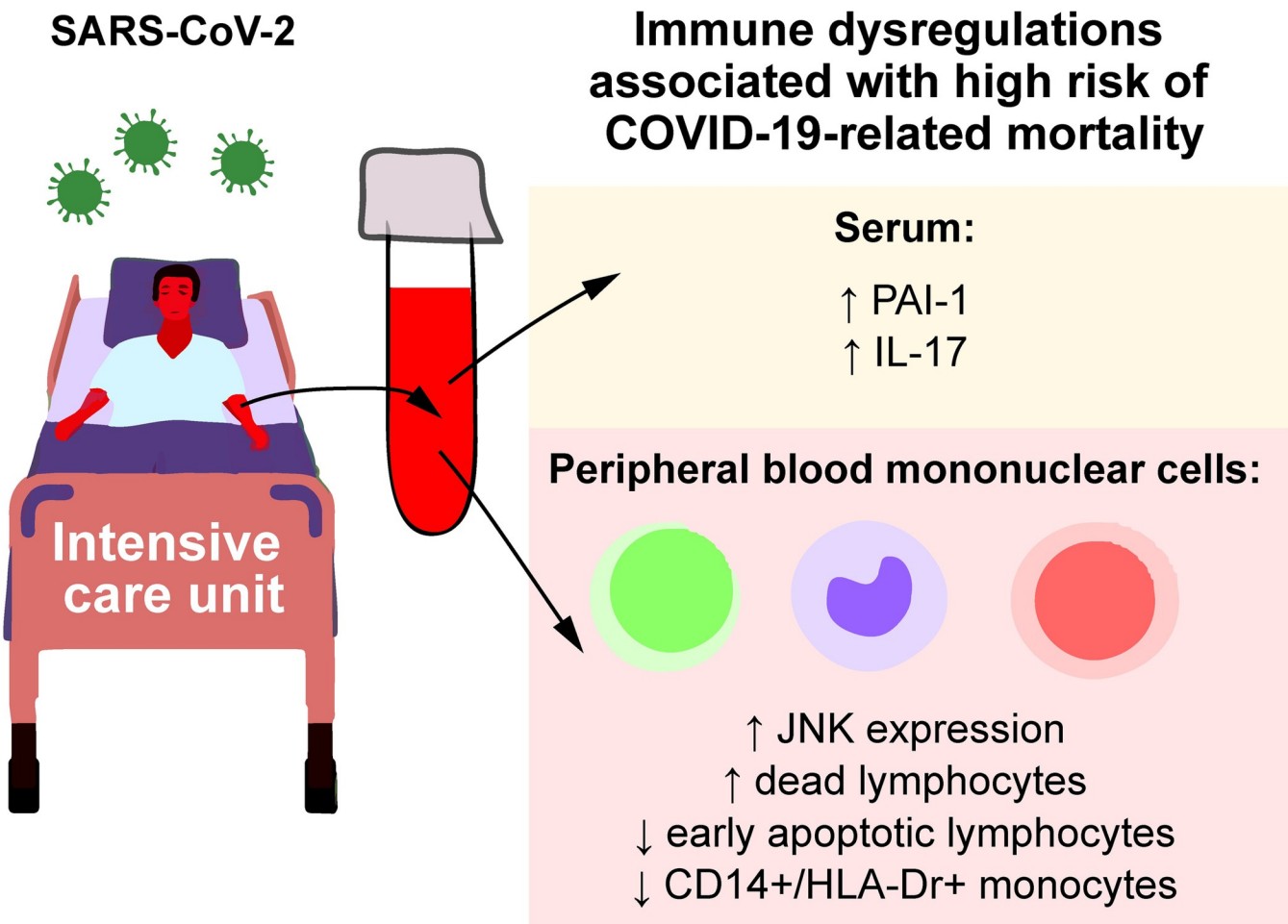

**Fig 4. Novel factors that suggest a high risk of COVID-19-related death overtime in patients with severe COVID-19 admitted to ICU.**

Our study has some limitations. First, the cohort might not be fully representative as the study was a single center one and only the patients admitted to ICU were included. Therefore, a relatively high mortality rate was detected, which cannot reflect the actual COVID-19 mortality in a population. Second, our results might be limited by the sample size. Third, the outcomes were evaluated at the end of the follow-up period instead of a fixed time period during the course of the disease. Fourth, not all laboratory tests were done in all patients, however, the data were missing completely at random. Hence, the interpretation of our results might be limited.

However, we believe that our study uncovers some important tendencies that should be further considered with the relation to COVID-19 mortality. As far as we are aware, this is the first retrospective cohort study among patients with COVID-19 in Russia that evaluated the biochemical parameters in blood serum different from those that are routinely used in healthcare practice. The obtained results may provide a rationale for testing these new parameters as novel markers of COVID-19 severity and point to potential targets for therapeutical intervention in quest to improve the outcomes.

## Supporting information

**S1 Table. The accepted reference values of the routine blood tests parameters.**
(DOCX)

**S2 Table. Categorization of biochemical parameters.** Reference group is highlighted in bold. MFI–Median Fluorescence Intensity.
(DOCX)

**S3 Table. Univariable Cox proportional hazards survival analysis models.** HR–hazard ratio, CI–confidence interval, Inf–infinite. $^*$–$P < 0.05$, $^{**}$–$P < 0.01$.
(DOCX)

**S1 Fig. The flow cytometry analysis.** A–Representative flow cytometry graphs showing the gating strategy used to assess lymphocyte apoptosis. Forward scatter (FSC) vs. side scatter (SSC) plot applied to gate the lymphocyte cell population and remove the debris. Annexin-FITC-channel vs. 7AAD-channel plot used to gate early (Annexin V+/7AAD-) and late (Annexin V+/7AAD+) apoptotic cells in lymphocyte population. B–Representative flow cytometry graphs showing the gating strategy used to evaluate the percentage of CD14+/HLA-Dr+ cells in monocyte cell population from the whole blood. Isotypic non-immune IgG (IgG1 Mouse-FITC Isotype Control, A07795, Beckman Coulter) was applied as a control.
(TIF)

**S2 Fig. Kaplan-Meier survival curves for demographic, other clinical and routine blood test parameters.** The fraction of survival is expressed on the y-axis, while time (days) of the observation period is expressed on the x-axis. Vertical traits indicate censored data (hospital discharge). *P* values of log-rank tests are indicated for each graph. Reference group is shown in red.
(TIF)

**S3 Fig. Kaplan-Meier survival curves for other biochemical test parameters.** The fraction of survival is expressed on the y-axis, while time (days) of the observation period is expressed on the x-axis. Vertical traits indicate censored data (hospital discharge). *P* values of log-rank tests are indicated for each graph. Reference group is shown in red.
(TIF)

## Author Contributions

**Conceptualization:** Kseniya Rubina, Anatoliy Evseev, Konstantin Popugaev, Sergey Petrikov.

**Data curation:** Anna Shmakova, Aslan Shabanov, Yulii Andreev, Natalia Borovkova, Vladimir Kulabukhov, Anatoliy Evseev, Konstantin Popugaev, Ekaterina Semina.

**Investigation:** Aslan Shabanov, Yulii Andreev, Natalia Borovkova, Vladimir Kulabukhov, Anatoliy Evseev, Konstantin Popugaev, Sergey Petrikov.

**Methodology:** Anna Shmakova, Aslan Shabanov, Yulii Andreev, Natalia Borovkova, Vladimir Kulabukhov, Anatoliy Evseev, Konstantin Popugaev.

**Resources:** Aslan Shabanov, Yulii Andreev, Vladimir Kulabukhov.

**Software:** Anna Shmakova.

**Supervision:** Kseniya Rubina, Sergey Petrikov, Ekaterina Semina.

**Validation:** Ekaterina Semina.

**Writing – original draft:** Kseniya Rubina.

**Writing – review & editing:** Kseniya Rubina, Ekaterina Semina.

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
