## [Decision Letter · Decision Letter 0]

5 Oct 2021

PONE-D-21-19114Novel prognostic determinants of COVID-19-related mortality: a pilot study on severely-ill patients in RussiaPLOS ONE

Dear Dr. Rubina,

Thank you for submitting your manuscript to PLOS ONE. After careful consideration, we feel that it has merit but does not fully meet PLOS ONE’s publication criteria as it currently stands. Therefore, we invite you to submit a revised version of the manuscript that addresses the points raised during the review process.

We look forward to receiving your revised manuscript.

Kind regards,

Jennifer A. Hirst, DPhil

Academic Editor

PLOS ONE

Journal Requirements:

3. PLOS requires an ORCID iD for the corresponding author in Editorial Manager on papers submitted after December 6th, 2016. Please ensure that you have an ORCID iD and that it is validated in Editorial Manager. To do this, go to ‘Update my Information’ (in the upper left-hand corner of the main menu), and click on the Fetch/Validate link next to the ORCID field. This will take you to the ORCID site and allow you to create a new iD or authenticate a pre-existing iD in Editorial Manager. Please see the following video for instructions on linking an ORCID iD to your Editorial Manager account: https://www.youtube.com/watch?v=_xcclfuvtxQ.

Additional Editor Comments (if provided):

This pilot study in 52 patients who had severe covid, determined prognostic factors associated with mortality using survival analysis (Cox and Kaplan Meyer)

There are some reporting issues with this manuscript which need to be revised before it is suitable for publication.

1. How were cut-offs for the biochemical parameters analysed in the Kaplan Meyer plots decided? Please clarify this in the methods and state whether these cut-offs had been pre-specified.

2. More details need to be provided in the methods on how variables were selected for inclusion in the multivariate Cox model in Table 3.

3. Figure S1 is not referred to anywhere in the text, should the text on page 5 refer to Figures S1 and S2 together?

4. The graphical abstract is not properly visible and is not referred to in the text. It has therefore not been reviewed. Please clarify what this is and whether it is necessary to the manuscript.

Reviewers' comments:

Reviewer's Responses to Questions

**Comments to the Author**

1. Is the manuscript technically sound, and do the data support the conclusions?

Reviewer #1: Yes

2. Has the statistical analysis been performed appropriately and rigorously? 

Reviewer #1: Yes

3. Have the authors made all data underlying the findings in their manuscript fully available?

Reviewer #1: Yes

4. Is the manuscript presented in an intelligible fashion and written in standard English?

Reviewer #1: Yes

5. Review Comments to the Author

Reviewer #1: This article focused on laboratory immunological parameters in blood serum as novel prognostic determinants for COVID-19 mortality and provided in depth discussion on their potential mechanism and clinical significance as disease severity predictors and potential therapeutical targets. There are several major/minor points to be addressed.

1. The prognostic value of the 6 laboratory immunological parameters remains unclear. There are differences between “relevant (associated with)” and “decisive (determinant)”. In the survival analysis using the Kaplan-Meier method, these 6 laboratory parameters are associated with a high risk of COVID-19 related mortality. However, in the multivariable Cox proportional hazards survival analysis model, 4 of the above-mentioned parameters lost statistical significance. Besides, clinical and routine blood covariates which are of statistical significance should be mentioned in the conclusion, instead of stating that “Gender … several parameters of routine blood tests were not statistically significant”.

2. The inaccessibility of these immunological laboratory tests limits their use in the real-world setting. I wonder their superiority compared with routine blood tests, which has been verified in a considerably amount of published COVID-19 researches in large patient cohorts.

3. Please provide detailed data in the alive group as an independent column between the “All patients” and “Deaths” column in Table 1 and make comparisons (provide P value) between the alive and dead group.

4. Information on baseline patient characteristics in Table 1 is limited. Please provide data on underlying comorbidities, reasons for ICU admission, complications during hospital stay, major treatment applied if possible.

5. Please explain why inflammatory cytokines such as IL-2,4,6,10, IFN γ were not tested.

6. It is suggested to present detailed data of clinical biochemical routine blood test result and laboratory blood findings as a supplementary material.

7. The reference of Chest CT severity score should be cited in the “Outcome and exposures” paragraph.

8. There are two Figure S1 in the Supplementary material and corresponding misleading quote in the manuscript. Please revise them.

6. PLOS authors have the option to publish the peer review history of their article (what does this mean?). If published, this will include your full peer review and any attached files.

Reviewer #1: No

---

## [Author Response · Author response to Decision Letter 0]

5 Nov 2021

Dear Dr. Hirst,

We would like to thank you and the Editorial board for the opportunity to resubmit the revised manuscript to PLOS ONE journal. Please, find the cover letter with the list of changes and rebuttal concerning each point.

The authors would like to thank the Reviewers for their attentive reading and unbiased reviewing of our manuscript, as well as for their comments and discussions. The comments put forward by the Reviewers were taken into the account and the corresponding changes were made. We believe this has strengthened the manuscript and we hope the revised version of it would be accepted for the publication.

Editor’s comments:

We revised the article accordingly.

We performed the corresponding changes.

According to data availability statement, the full data table with all the clinical and biochemical anonymous results is now available at (https://osf.io/tzn89/?view_only=2c70847e06a34fb0bffb9f61246c60db, 

doi: 10.17605/OSF.IO/TZN89) with free access.

3. PLOS requires an ORCID iD for the corresponding author in Editorial Manager on papers submitted after December 6th, 2016. Please ensure that you have an ORCID iD and that it is validated in Editorial Manager. To do this, go to ‘Update my Information’ (in the upper left-hand corner of the main menu), and click on the Fetch/Validate link next to the ORCID field. This will take you to the ORCID site and allow you to create a new iD or authenticate a pre-existing iD in Editorial Manager. Please see the following video for instructions on linking an ORCID iD to your Editorial Manager account: https://www.youtube.com/watch?v=_xcclfuvtxQ.

We performed the corresponding changes.

Additional Editor Comments (if provided):

This pilot study in 52 patients who had severe covid, determined prognostic factors associated with mortality using survival analysis (Cox and Kaplan Meyer). There are some reporting issues with this manuscript which need to be revised before it is suitable for publication.

1. How were cut-offs for the biochemical parameters analysed in the Kaplan Meyer plots decided? Please clarify this in the methods and state whether these cut-offs had been pre-specified.

The cut-offs are now described in detail in Materials and Methods section:

“Routine blood test parameters were organized into three groups relying on the pre-specified reference range accepted in Moscow hospital laboratories, including the clinical laboratory of N.V. Sklifosovsky Research Institute for Emergency Medicine: lower, within the reference range, or higher than the reference range.”

To ensure an adequate repartition in the groups, the categorization of the biochemical parameters in blood serum and leukocytes was performed basing on the pre-specified reference values provided by the manufacturer in the kit instructions, literature data and evaluation of the histogram distribution data. The cut-off values for biochemical serum and leukocyte parameters are presented in S2 Table.”

2. More details need to be provided in the methods on how variables were selected for inclusion in the multivariate Cox model in Table 3.

We provided the details in Materials and Methods section:

“The inclusion of variables in the final models were based on the following criteria: 1) significance in Kaplan-Meier and/or univariable analyses; 2) association with survival according to previous studies [9-14]; 3) clinical knowledge.”

3. Figure S1 is not referred to anywhere in the text, should the text on page 5 refer to Figures S1 and S2 together?

We thank the editor for this remark. The numbering of supplementary figures has been corrected.

4. The graphical abstract is not properly visible and is not referred to in the text. It has therefore not been reviewed. Please clarify what this is and whether it is necessary to the manuscript.

We performed the corresponding changes and the graphical abstract has been inserted in the manuscript. 

Reviewer’s comments:

1. The prognostic value of the 6 laboratory immunological parameters remains unclear. There are differences between “relevant (associated with)” and “decisive (determinant)”. In the survival analysis using the Kaplan-Meier method, these 6 laboratory parameters are associated with a high risk of COVID-19 related mortality. However, in the multivariable Cox proportional hazards survival analysis model, 4 of the above-mentioned parameters lost statistical significance. Besides, clinical and routine blood covariates which are of statistical significance should be mentioned in the conclusion, instead of stating that “Gender … several parameters of routine blood tests were not statistically significant”.

We restructured and re-wrote the Discussion section, paying more attention to the description of the results. We now step by step describe the significant clinical, routine blood covariates and laboratory results, mentioning which of them remained significant as independent risk factors in multivariable model. We also removed the phrase “Gender … several parameters of routine blood tests were not statistically significant” and added this phrase concerning the clinical and routine blood covariates, which were of statistical significance to the Conclusions.

“Among routine parameters, chest CT score, ventilation, increased WBC and neutrophil count, increased serum levels of globulins, urea, CK, augmented D-dimer and INR, decreased lymphocyte count and decreased prothrombin are risk factors for COVID-19-related mortality. The analysis of other factors, not routinely used in clinical practice, suggests a high risk of COVID-19-related death overtime in patients with elevated dead lymphocyte counts, decreased early apoptosis of lymphocytes, reduced fraction of CD14+/HLA-Dr+ monocytes, increased JNK expression in PBMCs, elevated IL-17 and decreased PAI-1 serum levels.”

2. The inaccessibility of these immunological laboratory tests limits their use in the real-world setting. I wonder their superiority compared with routine blood tests, which has been verified in a considerably amount of published COVID-19 researches in large patient cohorts.

Although the proposed laboratory biochemical and immunological tests are quite new and not routinely used, they are relatively easy-to-do and can be introduced into clinical practice. Moreover, the poor survival rates of high-risk patients warrant investigation into novel factors implicated in COVID-19 pathogenesis, as well as diagnostic and treatment options.

3. Please provide detailed data in the alive group as an independent column between the “All patients” and “Deaths” column in Table 1 and make comparisons (provide P value) between the alive and dead group.

We thank the Reviewer for this important suggestion. We updated the required information in Table 1 regarding the “Alive” group. The statistical comparison between survival and non- survival groups was also performed and P values has been provided.

4. Information on baseline patient characteristics in Table 1 is limited. Please provide data on underlying comorbidities, reasons for ICU admission, complications during hospital stay, major treatment applied if possible.

We thank the Reviewer for this important suggestion. We updated the required information. The major groups of underlying comorbidities (Hypertension, Congestive heart failure, Ischemic heart disease, Chronic kidney disease, Diabetes, Malignancy, Cerebrovascular disease) are now provided in Table 1. Major complications during hospital stay (Hydrothorax, Intoxication, sepsis, multiple organ failure, Acute respiratory distress syndrome, Hemorrhagic shock, Acute myopericarditis, Pseudomembranous colitis, Deep vein thrombosis of the lower extremities) were added to Table1. Applied treatments were added to Table 1 and described in detail in Materials and Methods section “Study design and clinical workflow”. The statistical comparison between survival and non- survival groups regarding these factors has been carried out and P values are provided. Reasons for ICU admission are now described in Materials and Methods section “Study design and clinical workflow”: 

“Reasons for ICU admission included the functional state deterioration with symptoms such as fever, cough, sore throat, signs of pneumonia and respiratory distress in case a patient was hospitalized from home (51 patients) or transferred from another hospital (1 patient)”

The anonymized data on each patient about their underlying comorbidities, reasons for ICU admission, complications during hospital stay, major treatment applied are also presented in a recapitulative table freely available at https://osf.io/tzn89/?view_only=2c70847e06a34fb0bffb9f61246c60db.

5. Please explain why inflammatory cytokines such as IL-2,4,6,10, IFN γ were not tested.

Since the present study aimed to discover novel prognostic factors, we examined several potentially perspective cytokines, such as IL-17, IL-1α, TGFβ, TNFα and assessed their prognostic significance in our cohort. The choice was based on the previously published literature data at the time of writing.

The cytokines Il-2, IL-6, IL-10, IFN-γ, mentioned by the Reviewer, have been already extensively studied in relation to COVID-19 severity and mortality rates as discussed in the Discussion section:

“A potentially practical opportunity was proposed by the sustained elevated production of IL-6 [2,4,11,17–19,21,23–25,40], IL-1, IL-2R, IL-8, IL-10 [2,11] and some other blood circulating factors, whose early evaluation anticipated disease progression and duration of hospital stay of COVID-19 patients [41]. Such interleukins as IL-1β, IL-6, IL-17A, TNFα, and monocyte chemoattractant peptide (MCP)-1 were identified in patients with severe COVID-19 [42]. Huang et al. [43] demonstrated the upregulated levels of IL-1β, IL-6, IL-8, IL-17, interferon γ (IFN-γ), TNFα in patients with COVID-19 as compared to healthy donors. The same study revealed increased plasma concentrations of MCP-1 and TNFα in patients with COVID-19 admitted to the ICU. Similarly, Chen and co-authors [6] reported the correlation between high concentrations of plasma IL-6 and TNFα and the severe course of COVID-19. In contrast, in the study by Kang and co-authors these cytokines, including IL-1β, IL-12p40, and IL-17 were undetectable in patients with severe COVID-19 [44]. Several clinical trials investigating interleukin inhibitors (IL-1 inhibitor anakinra, and IL-6 inhibitors tocilizumab or sarilumab) revealed that IL-1 inhibition, but not IL-6 inhibition, was associated with a significant reduction of mortality in patients admitted to hospital with COVID-19, ARDS (acute respiratory distress syndrome), and hyperinflammation [45].”

Several studies that analyzed IL-4 found no correlation with disease severity or mortality (PMID: 32501293, PMID: 31986264, PMID: 33914789), we therefore didn’t mention IL-4 in the Discussion.

6. It is suggested to present detailed data of clinical biochemical routine blood test result and laboratory blood findings as a supplementary material.

According to data availability statement, the table including the full data with all the clinical and biochemical anonymous results is now available at https://osf.io/tzn89/?view_only=2c70847e06a34fb0bffb9f61246c60db.

7. The reference of Chest CT severity score should be cited in the “Outcome and exposures” paragraph.

We performed the corresponding changes and cited the appropriate reference in the mentioned paragraph:

“Chest CT score was grouped as 1-2 or 3-4 based on semi-quantitative CT scoring ranging from 0 to 4, which was evaluated according to the protocol recommended by Moscow Healthcare Department adapted from International Protocols and enriched with local experience [58].”

8. There are two Figure S1 in the Supplementary material and corresponding misleading quote in the manuscript. Please revise them.

We performed the corresponding changes and the numbering of supplementary figures has been corrected.

Sincerely,

Ksenia A. Rubina, Doctor of Sciences, PhD,

Head of Laboratory of Morphogenesis and Tissue Reparation, Faculty of Medicine, Lomonosov Moscow State University,

Moscow, Russian Federation

Mobile tel. +7 (903) 728-88-21

Email: kseniiarubina1971@gmail.com

---

## [Decision Letter · Decision Letter 1]

4 Jan 2022

PONE-D-21-19114R1Novel prognostic determinants of COVID-19-related mortality: a pilot study on severely-ill patients in RussiaPLOS ONE

Dear Dr. Rubina,

Thank you for submitting your manuscript to PLOS ONE. After careful consideration, we feel that it has merit but does not fully meet PLOS ONE’s publication criteria as it currently stands. Therefore, we invite you to submit a revised version of the manuscript that addresses the points raised during the review process.

Please make the final edit as requested by the Reviewer.

We look forward to receiving your revised manuscript.

Kind regards,

Jennifer A. Hirst, DPhil

Academic Editor

PLOS ONE

Reviewers' comments:

Reviewer's Responses to Questions

**Comments to the Author**

1. If the authors have adequately addressed your comments raised in a previous round of review and you feel that this manuscript is now acceptable for publication, you may indicate that here to bypass the “Comments to the Author” section, enter your conflict of interest statement in the “Confidential to Editor” section, and submit your "Accept" recommendation.

Reviewer #1: (No Response)

2. Is the manuscript technically sound, and do the data support the conclusions?

Reviewer #1: Yes

3. Has the statistical analysis been performed appropriately and rigorously? 

Reviewer #1: Yes

4. Have the authors made all data underlying the findings in their manuscript fully available?

Reviewer #1: Yes

5. Is the manuscript presented in an intelligible fashion and written in standard English?

Reviewer #1: Yes

6. Review Comments to the Author

Reviewer #1: The discussion section in the revised manuscript seems not have been restructured and rewrote according to the authors' response to my first comment, and the sentence "Gender … several parameters of routine blood tests were not statistically significant" is not removed.

7. PLOS authors have the option to publish the peer review history of their article (what does this mean?). If published, this will include your full peer review and any attached files.

Reviewer #1: No

---

## [Author Response · Author response to Decision Letter 1]

17 Jan 2022

Reviewer #1: The discussion section in the revised manuscript seems not have been restructured and rewrote according to the authors' response to my first comment, and the sentence "Gender … several parameters of routine blood tests were not statistically significant" is not removed.

We restructured the Discussion section to the best of our understanding of the Reviewer’s comments. 

Concerning the laboratory immunological parameters, we underscored the importance of IL-17 and PAI-1 measurements that were significantly associated with survival by Kaplan-Meier and Cox proportional hazards estimates. We also removed the phrase "Gender … several parameters of routine blood tests were not statistically significant" from the discussion section.

Prognostic value of various parameters is canonically evaluated using both, univariable Kaplan-Meier and multivariable Cox regression survival analyses (i.e. PMID: 34675225, PMID: 31901133). Our study has certain limitations as mentioned in the last part of the discussion sections. Due to the total sample size and the event number (deaths) of the study, we considered important to discuss the parameters that are significant by Kaplan-Meier survival analysis and non-significant by Cox proportional hazards survival analysis model. Multivariable analysis (Cox model) was used to reveal independent risk factors, however due to the current experimental settings and the limited statistical power of Cox model this can result in non-significance of the parameters identified by Kaplan-Meier estimates. The aim of the present study was to highlight the potentially relevant prognostic factors that can be further investigated in detail with larger patient cohorts defining practical issues and future perspective. 

We have been confused by the following Reviewer’s comment concerning “relevant” and “decisive” factors, as we were unable to find these terms in Statistical Analysis literature. However, we revised the conclusions in a more accurate and correct way as recommended by the Reviewer.

With best regards,

Ksenia A. Rubina, Doctor of Sciences, PhD,

Head of Laboratory of Morphogenesis and Tissue Reparation, Faculty of Medicine, Lomonosov Moscow State University,

Moscow, Russian Federation

Mobile tel. +7 (903) 728-88-21

Email: kseniiarubina1971@gmail.com

---

## [Decision Letter · Decision Letter 2]

3 Feb 2022

Novel prognostic determinants of COVID-19-related mortality: a pilot study on severely-ill patients in Russia

PONE-D-21-19114R2

Dear Dr. Rubina,

We’re pleased to inform you that your manuscript has been judged scientifically suitable for publication and will be formally accepted for publication once it meets all outstanding technical requirements.

Kind regards,

Jennifer A. Hirst, DPhil

Academic Editor

PLOS ONE

Additional Editor Comments (optional):

Reviewers' comments:

Reviewer's Responses to Questions

**Comments to the Author**

1. If the authors have adequately addressed your comments raised in a previous round of review and you feel that this manuscript is now acceptable for publication, you may indicate that here to bypass the “Comments to the Author” section, enter your conflict of interest statement in the “Confidential to Editor” section, and submit your "Accept" recommendation.

Reviewer #1: All comments have been addressed

2. Is the manuscript technically sound, and do the data support the conclusions?

Reviewer #1: Yes

3. Has the statistical analysis been performed appropriately and rigorously? 

Reviewer #1: Yes

4. Have the authors made all data underlying the findings in their manuscript fully available?

Reviewer #1: Yes

5. Is the manuscript presented in an intelligible fashion and written in standard English?

Reviewer #1: Yes

6. Review Comments to the Author

Reviewer #1: The authors have adequately addressed my comments raised and this manuscript is now acceptable for publication.

7. PLOS authors have the option to publish the peer review history of their article (what does this mean?). If published, this will include your full peer review and any attached files.

Reviewer #1: No

---

## [Editor Report · Acceptance letter]

17 Feb 2022

PONE-D-21-19114R2 

Novel prognostic determinants of COVID-19-related mortality: a pilot study on severely-ill patients in Russia 

Dear Dr. Rubina:

I'm pleased to inform you that your manuscript has been deemed suitable for publication in PLOS ONE. Congratulations! Your manuscript is now with our production department. 

Kind regards, 

on behalf of

Dr. Jennifer A. Hirst 

Academic Editor

PLOS ONE